# The Cooperation Regulation of Antioxidative System and Hormone Contents on Physiological Responses of *Wedelia trilobata* and *Wedelia chinensis* under Simulated Drought Environment

**DOI:** 10.3390/plants13040472

**Published:** 2024-02-07

**Authors:** Ping Huang, Zhiwei Xu, Weijie He, Hong Yang, Bin Li, Wendian Ding, Yuze Lei, Adeel Abbas, Rashida Hameed, Congyan Wang, Jianfan Sun, Daolin Du

**Affiliations:** 1Institute of Environment and Ecology, School of Environment and Safety Engineering, Jiangsu University, Zhenjiang 212013, China; 2School of Emergency Management, Jiangsu University, Zhenjiang 212013, China; 3School of Environment and Safety Engineering, Jiangsu University, Zhenjiang 212013, China

**Keywords:** drought stress effects, metabolite disruption, antioxidant enzymes, adaptation strategies, climate changes

## Abstract

Drought-induced metabolic dysregulation significantly enhances the production of reactive oxygen species (ROS), which, in turn, exerts a substantial influence on the oxidation–reduction regulatory status of cells. These ROS, under conditions of drought stress, become highly reactive entities capable of targeting various plant organelles, metabolites, and molecules. Consequently, disruption affects a wide array of metabolic pathways and eventually leads to the demise of the cells. Given this understanding, this study aimed to investigate the effects of different drought stress levels on the growth and development of the invasive weed *Wedelia trilobata* and its co-responding native counterpart *Wedelia chinensis*. Both plants evolved their defense mechanisms to increase their antioxidants and hormone contents to detoxify ROS to avoid oxidative damage. Still, the chlorophyll content fluctuated and increased in a polyethylene-glycol-simulated drought. The proline content also rose in the plants, but *W. chinensis* showed a significant negative correlation between proline and malondialdehyde in different plant parts. Thus, *W. trilobata* and *W. chinensis* exhibited diverse or unlike endogenous hormone regulation patterns under drought conditions. Meanwhile, *W. trilobata* and *W. chinensis* pointedly increased the content of indole acetic acid and gibberellic acid in a different drought stress environment. A positive correlation was found between endogenous hormones in other plant parts, including in the roots and leaves. Both simulated and natural drought conditions exerted a significant influence on both plant species, with *W. trilobata* displaying superior adaptation characterized by enhanced growth, bolstered antioxidant defense mechanisms, and heightened hormonal activities.

## 1. Introduction

Various climatological models proffer the prognostication that with the concomitant reduction in precipitation and escalating temperatures, instances of drought will become increasingly recurrent on a global scale [1]. The imposition of drought stress exerts deleterious effects on the trajectory of plant growth and development, constraining access to vital water resources, thereby impeding pivotal physiological processes, such as photosynthesis and respiration, ultimately encumbering the abundance and geographic dispersion of plant populations across the global landscape [2,3]. Moreover, it begets a diminution in cell turgidity, hinders the equilibrium of source–sink dynamics, and disrupts the homeostasis of metabolic processes [4]. The manifestations of drought encompass ostensible morphological and physiological alterations in plants, and their adaptability can transmute them into species possessing drought tolerance attributes, be they native or invasive [5]. Plants evince evolutionary adjustments in response to a plenitude of growth circumstances, and the extent of their tolerance to these vicissitudes exhibits a conspicuous variability [6]. The scrutiny of compromised germination, subterranean structures (i.e., roots), and the quantification of sundry physiological and biochemical constituents serve as instrumental metrics in the assessment of these disparities [7].

Under drought stress conditions, the addition of reactive oxygen species (ROS) triggers oxidative damage, resulting in substantial harm to cellular organelles [8]. This oxidative damage, which is a consequence of excessive ROS accumulation, extends its adverse effects to various cellular components, including the electron transport chain; lipid peroxidation within chloroplasts and mitochondria; and the deactivation of essential enzymatic activity, proteins, and nucleic acids. These detrimental processes ultimately culminate in a notable reduction in photosynthesis and, consequently, a decrease in crop yield [9]. To counteract the detrimental impact of ROS, plants employ a range of antioxidant enzymes, such as Ascorbate peroxidase (APX), catalase (CAT), dehydroascorbate reductase (DHAR), Glutathione reductase (G.R.), glutathione S-transferase (GST), peroxidase (POD), and superoxide dismutase (SOD). These enzymes assume a pivotal role in mitigating the accumulation of reactive oxygen species (ROS) [10,11]. More specifically, these enzymatic systems are instrumental in upholding a delicate equilibrium between the production and signaling of ROS, thereby reinforcing the plant’s capacity to endure adverse environmental conditions. Among the array of environmental challenges, drought stress looms as a significant concern, impacting a plethora of plant species [12,13]. In an investigation involving three commonly encountered Lantana variants exposed to constantly changing environment. The functions of heat-resistant antioxidant enzymes, specifically peroxidase (POD), superoxide dismutase (SOD), and catalase (CAT), in the elimination of ROS were explored, which provided insights that shed light on the potential significance of these enzymes in strengthening the plant’s resilience against adverse environmental factors [14].

Plants employ various strategies to adapt to fluctuating water stress conditions, including osmotic regulation, antioxidant enzyme activities, and a range of endogenous hormones [15]. These intrinsic plant hormones play essential roles in regulating physiological functions, enabling plants to effectively cope with drought stress [16]. Among these phytohormones, abscisic acid (ABA) and indole acetic acid (IAA) hold special significance as drought-responsive hormonal agents [17]. They exert authoritative control over numerous physiological and biochemical plant processes, ultimately enhancing drought resistance across various plant species. For instance, during drought stress, the concentration of abscisic acid (ABA) increases, leading to stomatal closure, reducing transpiration, and fortifying the plant’s resistance to drought [18]. Indole acetic acid (IAA), which is another key endogenous hormone, is prevalent in various plant tissues and organs, particularly those associated with robust growth metabolism in roots and leaves [19]. Its concentration can be adjusted in response to arid environments, although the mechanisms governing IAA regulation during drought are complex and vary between different plant species [20]. In the context of this research, the focus turned to *W. trilobata*, which is a low-lying invasive species known for its remarkable adaptability. Native to Central America, this plant successfully established itself in tropical and subtropical regions through invasions [14]. Introduced in the latter part of the twentieth century to China’s Guangdong, Guangxi, and Fujian provinces as an ornamental plant, its invasive nature was quickly identified, resulting in its classification as an invasive species on a global scale [21]. Prior research examining the impact of drought stress on oxidative arrangements and hormonal activities predominantly concentrated on cultivated plants [22]. However, there exists a notable gap in comprehensive research regarding the consequences of drought stress on the regulation of antioxidative systems, hormone levels, and the subsequent physiological responses in *W. trilobata* and *W. chinensis*. Consequently, to facilitate comparative analysis, we designed this study to investigate how different simulated drought conditions influence the regulation of antioxidative systems and hormone levels in *W. trilobata* and *W. chinensis* at the physiological level.

## 2. Results

Drought stress exerts a considerable impact on the growth and development of plants, often resulting in a noticeable reduction in biomass. Our findings indicate that both *W. trilobata* and *W. chinensis* experienced significant inhibition under PEG-induced simulated drought conditions, as well as natural drought conditions (Figure 1A,B). In comparison with the control group (C.K.), *W. trilobata* exhibited a substantial decrease in the number of leaves, plant height, root length, and lateral roots by 60%, 83.16%, 25.51%, and 68.87%, respectively.

*W. chinensis* showed a decrease of 70.59% in leaf number, 81.85% in plant height, 54.90% in root length, and 53.14% in lateral roots (Figure 1 and Figure 2). This shows that *W. trilobata* and *W. chinensis* produced different results regarding root development under the natural drought treatment: *W. trilobata* and *W. chinensis* had significantly fewer lateral roots in a natural drought environment, but compared with the control group (C.K.), *W. trilobata* increased its root length by 14.51% and 37.13%, and decreased its lateral root number by 42.23% and 60.01% (Figure 2), while *W. chinensis* showed no significant difference in root length (Figure 2). At the same time, compared with the C.K., *W. trilobata* and *W. chinensis* had reduced chlorophyll contents by 36.08% and 24.48% in natural drought. In comparison, the chlorophyll content of *W. chinensis* increased by 7.19% in PEG-simulated drought (Figure 3). The analysis of the aboveground and underground biomass and root–shoot ratio of *W. trilobata* and *W. chinensis* under simulated drought conditions provided insights into their clonal reproductive capacity and ecological adaptability (Figure 3). Both *W. trilobata* and *W. chinensis* exhibited a notable reduction in aboveground biomass when subjected to both PEG-induced simulated and natural drought treatments. However, a significant decrease in underground biomass was observed only in *W. chinensis*. In comparison with the control group (C.K.), both plant species displayed a significant increase in the root–shoot ratio (Figure 3). In comparison with the control group (C.K.), under both PEG-induced simulated drought and natural drought conditions, *W. trilobata* exhibited a obvisously decrease in aboveground biomass. The correlation analysis revealed significant positive correlations between the height of *W. trilobata* and the number of leaves, as well as the chlorophyll content. Additionally, the lateral root number showed significant positive correlations with the number of leaves, plant height, and chlorophyll content, while displaying a meaningful negative correlation with the root–shoot ratio (Table 1, *p* < 0.01). Similarly, the height of *W. chinensis* exhibited significant positive correlations with the number of leaves and root length. The lateral root number also showed significant positive correlations with the number of leaves and plant height, while exhibiting a significant negative correlation with the root–shoot ratio (Table 2).

Our research findings demonstrate that *W. trilobata* and *W. chinensis* exhibited significant increases in proline content by 47.46% and 29.31%, respectively, during natural drought compared with the control group (C.K.) (Figure 4). Additionally, the result of Figure 4 illustrates that *W. trilobata* exhibited higher levels of MDA (malondialdehyde) content than *W. chinensis* under normal growth conditions. However, when exposed to drought stress, *W. trilobata* experienced a substantial and dramatic increase in MDA content, reaching its peak during the PEG-simulated drought, with a 97.64% increase compared with the control group. The correlation analysis revealed a negative correlation between the proline and malondialdehyde (MDA) contents in both *W. trilobata* and *W. chinensis* with the number of leaves, plant height, and lateral roots under both the PEG-simulated and natural drought treatments. However, *W. chinensis* exhibited a significant negative correlation with the number of leaves, plant height, and lateral roots (Table 1 and Table 2).

### Endogenous Hormones Activities

The research found that *W. trilobata* showed significantly increased ABA contents in its leaves and roots under drought conditions compared with the C.K., and the ABA content in the *W. chinensis* leaves was the highest in natural drought, and the ABA content in the *W. chinensis* roots reached the maximum value during PEG-simulated drought (Figure 5). Meanwhile, *W. trilobata* and *W. chinensis* extensively increased the content of IAA in the stem in a drought environment. At the same time, the changes of IAA in leaves and roots showed different patterns (Table 3). *W. trilobata* significantly increased Gibberellic acid (GA) content in its leaves and stems during PEG-simulated drought, but *W. chinensis* significantly decreased the GA content in its roots and stems during PEG-simulated drought (Figure 6). Furthermore, *W. trilobata* significantly increased the GA content in its roots, and *W. chinensis* significantly decreased the GA content in its roots during natural drought. Comparative analysis of *W. trilobata* and *W. chinensis* hormone combination and ratio exhibited changes in the overground and underground parts. *W. trilobata* significantly reduced the ABA/IAA and ABA/GA ratios in the underground part of the plant but significantly increased the IAA/GA ratio in the overground part of the plant, and *W. chinensis* significantly decreased the ABA/IAA ratio in the underground part of the plant in both drought treatments. The ABA/GA ratio was significantly increased in the aboveground part of *W. chinensis* during PEG-simulated drought, and the IAA/GA ratio was significantly increased in the underground part of *W. chinensis* during natural drought. These results show that *W. trilobata* and *W. chinensis* exhibited different or unlike endogenous hormone regulation patterns under drought conditions. The correlation analysis discovered a significant positive correlation between the ABA/GA (abscisic acid/gibberellic acid) ratio and root length in *W. trilobata*, as well as a significant positive correlation between the IAA/GA (indole acetic acid/gibberellic acid) ratio and root length under drought conditions (Table 1). However, in *W. chinensis*, the ABA/IAA ratio showed a positive correlation with the number of leaves, plant height, and lateral roots, and a negative correlation with the root–shoot ratio and peroxidase activity. The ABA/GA ratio was meaningfully negatively correlated with the root length and positively correlated with the chlorophyll and malondialdehyde contents (Table 2).

## 3. Discussion

The plant growth index and biomass directly reflect the plant response to drought stress. Different scholars worldwide believe that Polyethylene glycol (PEG) as an osmotic regulator can be used to simulate drought [23]. PEG 6000 simulated drought stress closest to the soil drought level. When plants cope with drought stress, their growth were negatively affected, while their different physiological and photosynthesis factors as well as chlorophyll fluorescence were suppressed obviously [24]. different physiological factors and photosynthesis and chlorophyll fluorescence decreased Insufficient soil water directly leads to slow plant growth and reduced plant height and leaf area, which leads to significantly reduced photosynthesis in plants and a sharp decrease in plant biomass under natural drought stress [25]. The results of this study show that both plants appeared with different phenotypes, growth indexes, antioxidant systems, and the ratio of endogenous hormones under simulated drought conditions (Figure 1, Figure 2, Figure 5 and Figure 6). PEG-manufactured drought and natural drought had different influences on *W. trilobata* and *W. chinensis* in terms of their inhibitory effects on their growth (Figure 1 and Figure 2), which were mainly featured in the following aspects compared with natural drought: PEG-simulated drought had more significant inhibition on leaf number, plant height, root length, lateral root number and chlorophyll content of the two plants (Figure 1, Figure 2 and Figure 3). PEG-simulated drought can restrain the plant height, root length, and underground biomass of Coix lachryma-Jobi L seedlings [26]. In addition, PEG-simulated drought can affect the seed germination rate, germination index, and relative water content (RWC) of *Coix lachryma* seedlings [27].

In natural drought conditions, as the severity of drought increased, both plants exhibited a more pronounced decrease in the number of leaves, plant height, and chlorophyll content [28]. Conversely, the root–shoot ratio showed a more notable increase. *W. trilobata* demonstrated a significant increase in root length, suggesting its adaptation to drought by enhancing water absorption through an increased root length (Figure 2). In contrast, *W. chinensis* exhibited a significant decrease in lateral root number, with no significant change in root length (Figure 2). This study revealed that *W. trilobata* exhibited greater drought tolerance compared with *W. chinensis* under simulated drought conditions [21]. *W. trilobata* displayed enhanced adaptability and phenotypic plasticity when compared with other plant species. Furthermore, their investigation discovered a potential association between the invasion of *W. trilobata* and the richness of the soil fungal community [29].

Drought stress can disrupt the plant cell membrane system, and numerous studies demonstrated the accumulation of CAT, POD, MDA, and free proline in plants as a response to drought stress [30,31]. This accumulation serves to maintain the dynamic equilibrium of the antioxidant system. CAT plays a protective role in preventing the destruction of cell membranes, while POD aids in the elimination of excessive reactive oxygen species within the plant [32]. MDA serves as the product of lipid peroxidation in plant cells [33]. The accumulation capacity of free proline serves as an indicator of a plant’s drought resistance [34]. The substance under investigation interacts and forms cross-links with proteins localized on the cell membrane, thereby exerting a detrimental impact on the structure and functionality of the biofilm [35]. Furthermore, the constituents within this substance possess the capability to serve as indicators, reflecting the magnitude of cellular membrane damage [36]. The findings of this investigation demonstrate that under simulated drought conditions, both *W. trilobata* and *W. chinensis* exhibited a significant expansion in the levels of CAT, proline, and MDA within their leaves, while no significant alteration was observed in the POD content (Figure 4). Furthermore, within *W. chinensis* leaves, the CAT, proline, and MDA contents displayed a positive correlation with the severity of drought during natural drought conditions [37]. These observations suggest that *W. trilobata* and *W. chinensis* alleviate oxidative damage induced by drought stress by augmenting the concentration of free proline and modulating the activity of antioxidant enzymes [38]. Moreover, the rise in MDA content within *W. chinensis* was significantly greater than that in *W. trilobata* under the simulated drought conditions induced by PEG (Figure 4), implying that *W. chinensis* might encounter more pronounced oxidative damage to its cellular components under this particular treatment [39]. Additionally, it is worth noting that endogenous hormones play a remarkably fundamental role in regulating plant growth [40]. Different drought stress levels increase the ABA content (Table 3) in leaves of Populus euphratica, induce stomatal closure to reduce transpiration, and enhance the drought tolerance of *P. euphratica* [41]. ABA content in European rapeseed was positively correlated with the degree of drought and the most obvious increase in ABA content in leaves and root tips [42]. The results of this study show that compared with the control group, *W. trilobata* significantly increased the ABA content in its leaves and roots under drought conditions, while the ABA content in *W. chinensis’s* leaves was the highest during natural drought and in its roots during PEG drought [43]. This indicates that the response of different tissue parts of these two plants to the drought environment was different. *W. trilobata* and *W. chinensis* exhibited different endogenous hormones in the two simulated drought conditions, indicating different adaptation mechanisms: The hormone contents in aboveground parts and underground parts of *W. trilobata* (Table 3) followed ABA > IAA > GA during PEG-simulated drought [44]. The hormone content in aboveground parts and underground parts of *W. chinensis* followed IAA > ABA > GA during PEG-simulated drought; the distribution pattern of IAA > ABA > GA in the aboveground and underground of the two plants under natural drought was the same (Table 3) [45]. This result shows that *W. trilobata* and *W. chinensis* responded differently to the two different modes of drought, and these responses were related to the polar transport of endogenous hormones.

Plant hormones are trace organic compounds that regulate plant growth and development and usually play a critical function in plant life activities [46]. Abscisic acid (ABA) is ubiquitous in higher plants and involves many physiological activities, such as flowering time, seed dormancy, and fruit ripening [47]. Under drought stress, abscisic acid accumulates in leaves, stems, and roots to promote stomatal closure and water absorption by roots, thus improving the drought resistance of plants [48]. The chemical essence of Auxin (IAA) is indoleacetic acid, which is widely present in root tips, stem tips, and young leaves [49]. Under drought stress, the content of IAA in plants increases, and after rehydration, the content of IAA in plants shows a declining trend. Gibberellin (GA) is widely found in plants of more than 100 species, and its essence is a diterpenoid compound [50]. Gibberellins are primarily associated with promoting plant growth, including stem elongation, cell division, and seed germination [51]. These findings illuminate the intricate and species-specific responses of these plants to various drought stress conditions, underscoring the complexity of the hormonal regulation and growth patterns of *W. trilobata* and *W. chinensis.*

## 4. Materials and Methods

### 4.1. Experimental Materials and Design

For this study, specimens of *W. trilobata* were collected from Haikou, Hainan Province, China (20°02′45.97″ N, 110°11′38.39″ E), while *W. chinensis* samples were obtained from the greenhouse of the College of Environmental and Safety Engineering, Zhenjiang, Jiangsu University, China (32°12.02 N). Potted cultivation was employed, utilizing four distinct treatment conditions, with each treatment being replicated 12 times. The conditions of natural drought simulation were set as C.K. (watered every two days), mild drought (watered every four days), and moderate drought (watered every six days). The simulated drought concentration of PEG 6000 was 75 g/L. The plant growth substrate was a mixture of screened, washed, sterilized, and dried river sand and vermiculite (river sand:Vermiculite = 5:1) with same weight that weighed 300 g. The experiment required the choice of healthy branches and the same thickness of *W. trilobata* and *W. chinensis* regenerated stem pieces; each stem segment comprisedwith two leaves, and the selected stem segments were first soaked in sodium hypochlorite solution (5%) for surface disinfection and sterilization for 10 min, washed in sterile water 5 times, and then soaked in 0.1 times Hoagland nutrient solution with a randomized complete block design. When the new root grew at the end of the stem segment after 2–3 days, the cuttings were cut vertically in a square plastic pot, and three cuttings were cut in each pot. The cuttings were watered and soaked in the greenhouse of Jiangsu University; the temperature was 25 °C, the humidity was 60%, and the light cycle was 16 h/8 h (day/night). The drought treatment was applied after the appearance of new leaves, and a few roots appeared in the stem segment.

### 4.2. Phenotypic Growth Index

The aerial parts (leaves and stems), as well as the subterranean parts (roots), of both *W. trilobata* and *W. chinensis* were harvested for analysis. Various parameters, including the fresh weight, dry weight, plant height, leaf count, root length, and chlorophyll content, were measured. To prepare the samples, the freshly collected leaves, stems, and roots were carefully washed with clean water and subsequently subjected to an oven-drying process. The samples were initially exposed to a temperature of 105 °C for 10 min, followed by a subsequent drying period at 65 °C for 72 h. The sum of the dry leaf weight, dry stem weight, and root dry weight was the total biomass, where an analytical balance was used to measure them. The plants were taken photos with a digital camera, while the number of leaves, plant height, and root length were analyzed with Image J 1.38e (https://imagej.nih.gov/ij/) (accessed on 15 October 2021). Before harvesting the plants, the chlorophyll contents of the fresh leaves were measured using a portable chlorophyll meter. The measurement time was controlled to be from 9 a.m. to 10 a.m. The repeated measurements with different treatments should be made to ensure that the placement of the flowerpot was fixed as much as possible, as far apart as possible and took the average value for data analysis.

### 4.3. Biochemical Indices

The assessment of antioxidant defense systems encompassed measurements of the peroxidase (POD) and catalase (CAT) enzyme activities, as well as the determination of the proline and malondialdehyde (MDA) contents. The activity of POD was determined using the guaiacol method, while the activity of CAT was assessed using the UV absorption method. The proline content was determined using the acidic ninhydrin method, whereas the content of MDA was quantified using the thiobarbituric acid method [52].

### 4.4. Endogenous Hormone

The relative content of abscisic acid (ABA), indole acetic acid (IAA), and gibberellic acid (GA) in the leaves, stems, and roots of *W. trilobata* and *W. chinensis* was quantified using an ELISA kit. The endogenous hormone kit used for this analysis was procured from the Crop Chemical Control Laboratory at China Agricultural University [53].

A total of 0.5 g of plant tissue was stored at −80 °C, 2 mL extraction solution was added, and the mixture was ground under ice and transferred to a 10 mL tube. After rinsing the mortar with 2 mL of solution, it was transferred and refrigerated for 4 h. Following centrifugation at 3500 r/min for 8 min, the supernatant was collected. A total of 1 mL of solution was added to the precipitation, mixed, and refrigerated for 1 h before another centrifugation at 3500 r/min for 8 min. The supernatants were combined, the volume was recorded, and the residue was discarded. The supernatant underwent C-18 column processing with specific steps. The sample was then transferred to a 5 mL tube, concentrated, and dried under vacuum, followed by the addition of 1 mL of sample diluent to make up the volume. Next, for testing, the standard solution was diluted twice the standards were diluted in sequence twofold, with concentrations ranging from 0 to the maximum (50 ng/mL for GA, 100 ng/mL for IAA and ABA). Standards went in the first two columns of a 96-well enzyme plate, with 2 wells for each concentration. Test samples filled the remaining wells. A total of 50 μL of antibody was added to each well, mixed well, and incubated at 37 °C for 0.5 h. After washing the plate four times, 100 μL of secondary antibody was added to each well and incubated again at 37 °C for 0.5 h. The washing process was repeated, the substrate for color development was added, the reaction was terminated with sulfuric acid after color development, and colorimetry was performed using a microplate reader.

### 4.5. Data Analysis

SPSS (26.0) software was used to perform one-way or multivariate analysis of variance and correlation analysis on each factor. Multiple comparisons were conducted using Duncan’s test, with *p* < 0.05 indicating significant levels between treatments. The results of statistical analysis were plotted using origin 2018 (origin lab Co., Northhampton, MA, USA) software.

## 5. Conclusions

The research outcomes elucidate the physiological reactions of two closely related plant species, namely, *W. trilobata* and *W. chinensis*, to drought-induced stress. Notably, the investigation unveiled that under the influence of drought stress, *W. trilobata* manifested a substantial elevation in ABA levels within both its leaves and roots in comparison with the control cohort. Conversely, *W. chinensis* exhibited the highest ABA concentration in its leaves when subjected to natural drought, while the ABA content in its roots peaked under PEG-simulated drought. Moreover, within a drought milieu, both *W. trilobata* and *W. chinensis* showcased conspicuous increases in indole-3-acetic acid content in their stems. Intriguingly, the response of IAA in the leaves and roots displayed disparate patterns. For instance, *W. trilobata* had significantly augmented gibberellic acid content in its leaves and stems when exposed to PEG-simulated drought. In contrast, *W. chinensis* exhibited a noteworthy reduction in GA content within its roots and stems under identical conditions. The study also disclosed that amid natural drought circumstances, *W. trilobata* manifested a substantial upswing in GA content within its roots, whereas *W. chinensis* exhibited a marked downturn in GA content in its roots. These findings shed light on the intricate and species-specific reactions of these plants to diverse drought stress conditions, emphasizing the complexity of their hormonal regulation and growth patterns in response to environmental challenges.

## Figures and Tables

**Figure 1 plants-13-00472-f001:**
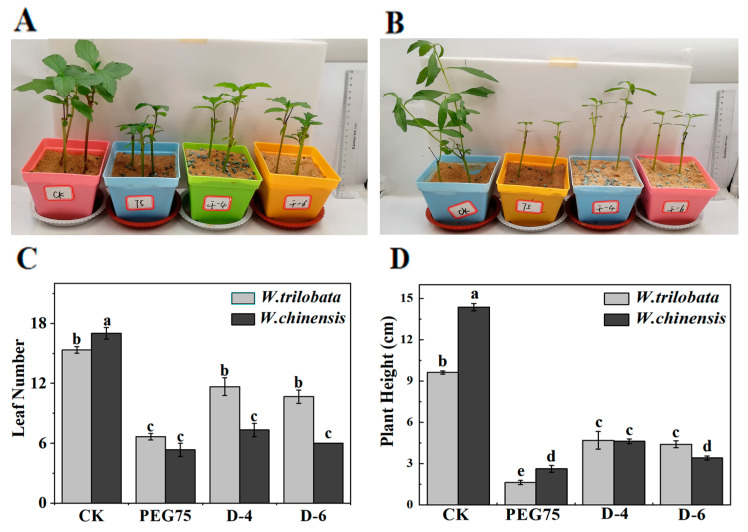
Effects of simulated drought stress on the growth of *W. trilobata* and *W. chinensis* (**A**). Growth phenotype of *W. trilobata*. (**B**) Growth phenotype of *W. chinensis*. (**C**) The effect of simulated drought stress on number of leaves of two plants. (**D**) The effect of simulated drought stress on plant height of two plants (mean ± SE, n = 3). Different lowercase letters in the figure indicate significant differences between different treatments (*p* < 0.05).

**Figure 2 plants-13-00472-f002:**
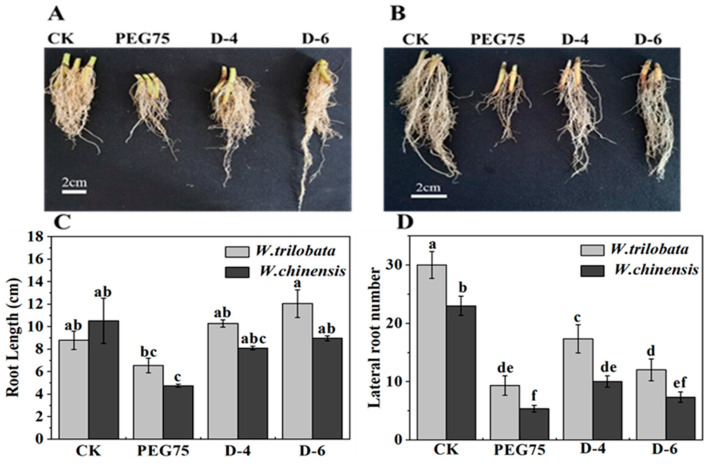
The influence of simulated drought stress on root growth in *W. trilobata* and *W. chinensis* was examined. (**A**) Root growth in *W. trilobata*. (**B**) Root growth in *W. chinensis*. (**C**) Root growth in both *W. trilobata* and *W. chinensis*. (**D**) Number of lateral roots in the two plants (mean ± standard error, n = 3). The lowercase letters in the figure denote significant differences observed between the treatments (*p* < 0.05). CK: the conditions of natural drought simulation, PEG75: Polyethylene glycol, D-4: water applied after four days, D-6: water applied after six days.

**Figure 3 plants-13-00472-f003:**
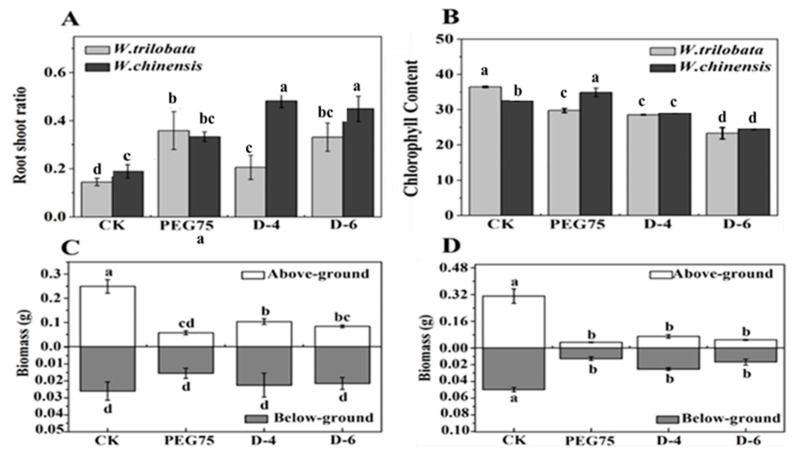
The effects of simulated drought stress on the root–shoot ratio (**A**), chlorophyll content (**B**), and biomass of *W. trilobata* (**C**), and biomass of *W. chinensis* (**D**) (mean ± S.E., n = 3). Different lowercase letters in the figure indicate significant differences between different treatments (*p* < 0.05).

**Figure 4 plants-13-00472-f004:**
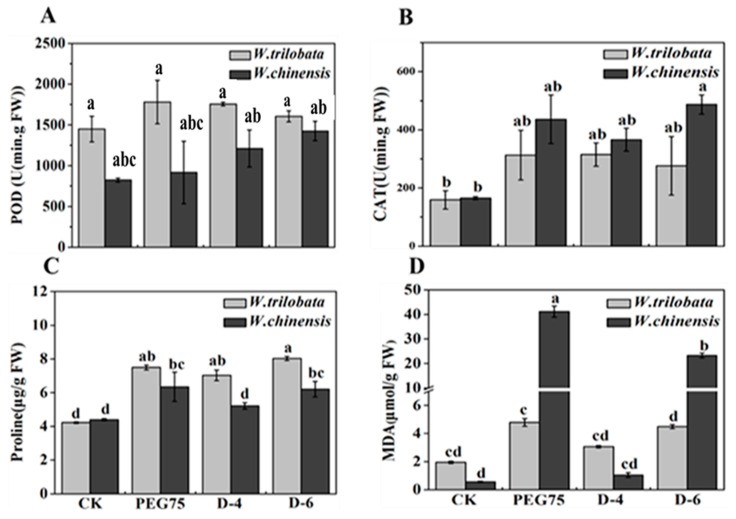
The impacts of simulated drought stress on peroxidase (**A**), catalase (**B**), proline (**C**), and malondialdehyde (**D**) in the leaves of the two plants were examined (mean ± standard error, n = 3). Distinct lowercase letters in the figure represent significant differences observed between the different treatments (*p* < 0.05).

**Figure 5 plants-13-00472-f005:**
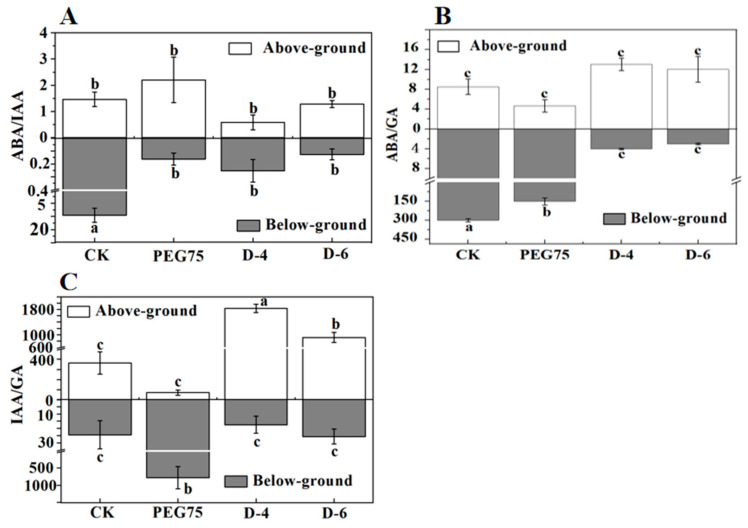
Effects of simulated drought stress on the ABA/IAA (**A**), ABA/GA (**B**), and IAA/GA (**C**) ratios in both the aboveground and underground parts of *W. trilobata*. The data presented represent the mean values with standard errors (mean ± standard error, n = 3). In the figure, lowercase letters are used to indicate significant differences observed between the treatments (*p* < 0.05).

**Figure 6 plants-13-00472-f006:**
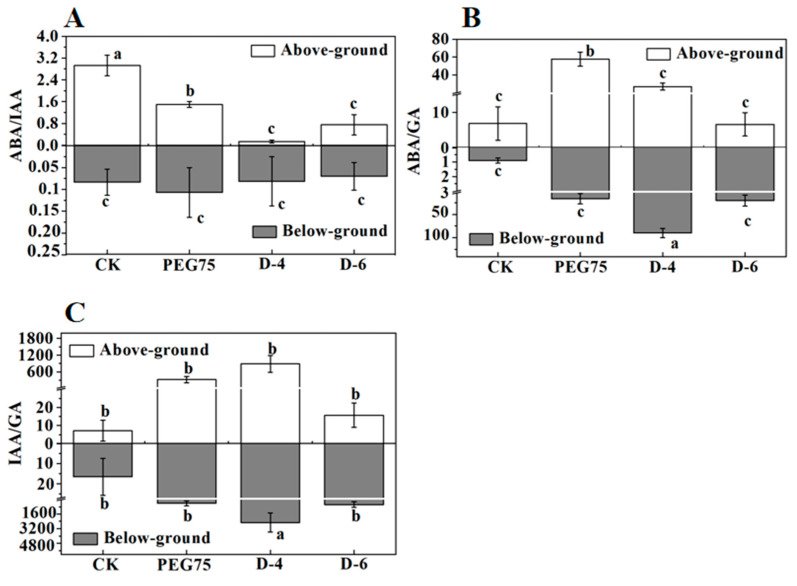
The impacts of simulated drought stress on the aboveground and underground ABA/IAA (**A**), ABA/GA (**B**), and IAA/GA (**C**) ratios in *W. chinensis* were examined (mean ± standard error, n = 3). Lowercase letters in the figure indicate significant differences between the treatments (*p* < 0.05).

**Table 1 plants-13-00472-t001:** Correlation analysis of the effects of simulated drought on growth index, antioxidant system, and endogenous hormone ratio of *W. trilobata*.

	LN	PH	RL	CH	RSR	LRN	Pro	MDA	POD	CAT	AI	AG	IG
**LN**	1	0.922 **	0.275	0.493	−0.562	0.881 **	−0.777 **	−0.851 **	−0.381	−0.484	−0.265	0.235	0.199
**PH**		1	0.192	0.615 *	−0.509	0.945 **	−0.856 **	−0.872 **	−0.468	−0.463	−0.191	0.303	0.038
**RL**			1	−0.559	0.035	0.034	0.174	−0.054	−0.298	0.029	−0.504	0.600 *	0.583 *
**CH**				1	−0.568	0.746 **	−0.875 **	−0.728 **	−0.260	−0.346	0.134	−0.274	−0.403
**RSR**					1	−0.589 *	0.591 *	0.699 *	0.001	0.124	−0.178	0.110	0.068
**LRN**						1	−0.944 **	−0.941 **	−0.349	−0.428	−0.128	0.253	0.007
**Pro**							1	0.868 **	0.390	0.478	−0.012	0.007	0.245
**MDA**								1	0.221	0.262	0.195	−0.218	−0.174
**POD**									1	0.629 *	0.519	−0.015	0.138
**CAT**										1	0.308	0.140	0.200
**Al**											1	−0.281	−0.584 *
**AG**												1	0.681 *
**IG**													1

LN: number of leaves, PH: plant height, RL: root length, CH: chlorophyll content, RSR: root–stem ratio, Pro: proline, MDA: malonaldehyde, POD: peroxidase, CAT: catalase, AI: ABA/IAA value, AG: ABA/GA value, IG: IAA/GA value. The symbol of “*” indicates significant correlations (*p* < 0.05) and “**” signifies exceedingly momentous correlation (*p* < 0.01).

**Table 2 plants-13-00472-t002:** Correlation analysis of the effects of simulated drought on growth index, antioxidant system, and endogenous hormone ratio of *W. chinensis*.

	LN	PH	RL	CH	RSR	LRN	Pro	MDA	POD	CAT	AI	AG	IG
**LN**	1	0.970 **	0.838 **	−0.572	0.001	0.909 *	−0.562	−0.466	−0.235	−0.769 **	−0.322	−0.464	−0.372
**PH**		1	0.799 **	−0.547	−0.080	0.919 **	−0.606 *	−0.515	−0.188	−0.742	−0.214	−0.525	−0.447
**RL**			1	−0.772 **	0.145	0.639 *	−0.358	−0.033	−0.278	−0.436	−0.531	−0.060	−0.011
**CH**				1	−0.193	−0.375	0.380	−0.309	0.530	0.134	0.493	−0.266	−0.183
**RSR**					1	0.026	0.322	0.430	0.394	0.361	−0.55 **	0.342	0.395
**LRN**						1	−0.432	−0.640 *	0.077	−0.685 *	−0.175	−0.643 *	−0.519
**Pro**							1	0.471	0.378	0.549	−0.120	0.508	0.642 *
**MDA**								1	−0.236	0.677 *	−0.464	0.977 **	0.892 **
**POD**									1	0.352	−0.010	−0.279	−0.176
**CAT**										1	−0.079	0.625 *	0.513
**Al**											1	−0.376	−0.416
**AG**												1	0.949 **
**IG**													1

LN: number of leaves, PH: plant height, RL: root length, CH: chlorophyll content, RSR: root–stem ratio, Pro: proline, MDA: malonaldehyde, POD: peroxidase, CAT: catalase, AI: ABA/IAA value, AG: ABA/GA value, IG: IAA/GA value. The symbol of “*” indicates significant correlations (*p* < 0.05) and “**” signifies exceedingly momentous correlation (*p* < 0.01).

**Table 3 plants-13-00472-t003:** The influence of varied drought conditions on hormone contents in *W. trilobata* and *W. chinensis*.

*W. trilobata*									
Treat-ment	ABA Content (ng/g FW)			IAA Content (ng/g Fw0P)			GAA Content (ng/g Fw)		
	Leaf	Stem	Root	Leaf	Stem	Root	Leaf	Stem	Root
CK	77.41 ± 12.59 d	24.34 ± 5.90 d	77.94 ± 10.2 c	62.84 ± 7.40 d	2234.83 ± 387.99 b	8.91 ± 1.50 d	35.94 ± 12.88 d	9.26 ± 6.48 b	0.59 ± 0.30 d
PEG75	318.36 ± 85.53 a	64.94 ± 11.25 a	327.05 ± 42.6 b	165.12 ± 29.10 c	1769.06 ± 213.94 c	2105.66 ± 182.3 b	264.77 ± 19.08 a	32.26 ± 10.09 a	3.96 ± 1.67 c
D-4	153.05 ± 34.82 c	46.83 ± 10.09 bc	379.40 ± 19.02 a	349.72 ± 23.0 a	9274.31 ± 129.98 a	1872.16 ± 358.9 bc	73.33 ± 16.02 bc	5.03 ± 0.28 b	114.71 ± 11.82 b
D-6	287.63 ± 29.42 b	54.07 ± 12.91 b	420.10 ± 62.78 a	227.20 ± 16.62 b	7002.72 ± 184.6 b	4138.90 ± 314.8 a	194.65 ± 27.6 b	8.70 ± 2.96 b	156.41 ± 20.60 a
*W. Chinensis*									
CK	155.52 ± 47.64 b	99.37 ± 2.78 b	174.28 ± 25.9 b	101.58 ± 9.48 d	68.98 ± 2.75 d	2710.41 ± 380.3 c	32.77 ± 14.8 c	55.05 ± 9.76 a	223.18 ± 31.4 a
PEG75	280.98 ± 50.67 a	235.12 ± 15.81 a	345.74 ± 77.5 a	215.80 ± 19.33 c	1223.51 ± 76.6 c	5087.35 ± 286.3 a	43.38 ± 18.01 c	4.40 ± 0.8 bc	92.93 ± 26.49 b
D-4	123.71± 21.89 bc	213.57 ± 33.9 a	114.71 ± 32.50 c	4325.41 ± 331.1 a	6937.79 ± 116.5 a	3521.73 ± 224.01 b	142.81± 75.8 b	10.23 ± 3.99 b	1.38 ± 0.52 d
D-6	302.76± 29.25 a	203.84 ± 52.2 a	337.05 ± 40.72 a	3037.17 ± 358.3 b	5485.87 ± 327.1 b	2129.40 ± 243.7 d	1007.08 ± 146.5 a	52.03 ± 13.01 a	30.34 ± 4.51 c

First four lines describe *W. trilobata* and second four lines describe *W. Chinensis*. Different lowercase letters in each column represent a distinct treatment condition and comparisons within the treatment groups. ABA: abscisic acid, IAA: indole acetic acid, GAA: gibberellic acid, CK: the conditions of natural drought simulation, PEG75: Polyethylene glycol, D-4: water applied after four days, D-6: water applied after six days.

## Data Availability

Data are available in the article.

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
