# Peer review of "The Cooperation Regulation of Antioxidative System and Hormone Contents on Physiological Responses of Wedelia trilobata and Wedelia chinensis under Simulated Drought Environment"

_plants, 2024, doi:10.3390/plants13040472_

Round 1
Reviewer 1 Report
Comments and Suggestions for Authors
I think that the manuscript entitled “Drought-Induced Metabolic Dysregulation: Antioxidative Responses and Hormone Regulation in Two Co-Occurring Plant Specie” is relevant to the Plants journal (section Plant Physiology and Metabolism/Special Issue: Abiotic Stress-Induced Secondary Metabolites Regulating Plant Metabolism) but needs many improvements before publication in this journal. In the present form, the manuscript is unsuitable for publication and needs revision.
Generally, the paper is relatively straightforward; however, the presentation and interpretation of results is not always clear. The Authors should improve the title. All sections of the manuscript need much more attention. The Authors should modify the beginning of the Introduction and more underline the purpose of the study. The Authors should also improve the Material and methods section - provide more information on extraction and enzyme estimation conditions, hormone measurements, provide the names of the equipment, and statistical analysis. The presentation and description of results should also be improved in the Results section (e.g., add units to Tables 1-2). Tables 3 and 4 contain the same values ​​and figures 4 and 5 are the same (the captions under them are also identical) - change it! The Authors should modify the beginning of the Discussion chapter. Also, the References list and citations must be more carefully checked. Perhaps the manuscript would gain some clarity if the text were corrected by a native English speaker. In addition, there are too many small mistakes in the text of manuscript (e.g., lines 4, 14, 25, 28, 29, 36, 56, 57, 59, 104, 116, 126, 147, 148, 153, 173, 218, 259, 266, 275, 319, 325, 343, 352, 361, 372, 395, 403, 409, 415, 438, 456, 457, 462, 476, 492, 495, 500, 508, 516, 538, 544, 552), that need to be corrected by Authors.
Author Response
"Please see the attachment."

Reviewer 2 Report
Comments and Suggestions for Authors
EVALUATION FORM
Title of manuscript:
Drought-Induced Metabolic Dysregulation: Antioxidative Re-2 sponses and Hormone Regulation in Two Co-Occurring Plant 3 Specie
Dear personalities,
The present manuscript could be located within the high-priority subjects of the PLANTS and in agreement with the publication criteria of this one, the manuscript could be considered as a ARTICLE SCIENTIST, because their experimental results belong to the one of an investigation made in the field of applied sciences and is directed to a hearing in the space studies of the AGRICULTURAL sector, with an approach of real application for the development of science and the technology and of the productive sector like is it the AGRICULTURAL sector.
Also the article is interesting because approaches an plants species which are of great importance in the world.
Unfortunately the document needs REVISIONS.
Recommendations:
- The manuscript is suitable for publication __________
- The manuscript is suitable for publication after revisions. _XXXX___
- The manuscript is not suitable for publication____________
The manuscript suggested a reviewed meticulously, taking care of in addition to the made ones in the manuscript, the following observations and considerations:
1.- In order to make agile the arbitration of the present manuscript, I worked directly in the sent file. On the matter it is indicated that I am sending for you the file and Evaluation form, which are will have to open and to activate the tool CONTROL OF CHANGES in the program of Word for Windows.
2.- All the observations and commentaries to the manuscript were made directly in the mentioned file, reason why the authors will have meticulously to review each one of the proposed changes, respective observations and/or commentaries.
3.- In each one of the sections that compose the present manuscript they stand out in “red” the pertinent corrections.
4.- Check the summary due the modifications made.
5.- In the INTRODUCTION section modifications were made.
6.- In the section of MATERIALS AND METHODS: In general, the authors must remember that in this section the information with clarity will have to be provided, with detail, because some colleague could be interested in verifying the experiments of this information. Nevertheless, details about methods commonly used and some materials will be able to be omitted.
7.- In the section of RESULTS In this section, a greater description of the results is required: this is good for your article. However they (the results) has to be writing in the same form that you explain in your metodology.
It is also considered that the authors can include a section of Conclusions where in summarized form more and clear and mainly objective they present the conclusions with respect to the objectives raised in the introduction section.
8.-The section of REFERENCES: SEE COMENTS

Comments on the Quality of English LanguageEVALUATION FORM
Title of manuscript:
Drought-Induced Metabolic Dysregulation: Antioxidative Re-2 sponses and Hormone Regulation in Two Co-Occurring Plant 3 Specie
Dear personalities,
The present manuscript could be located within the high-priority subjects of the PLANTS and in agreement with the publication criteria of this one, the manuscript could be considered as a ARTICLE SCIENTIST, because their experimental results belong to the one of an investigation made in the field of applied sciences and is directed to a hearing in the space studies of the AGRICULTURAL sector, with an approach of real application for the development of science and the technology and of the productive sector like is it the AGRICULTURAL sector.
Also the article is interesting because approaches an plants species which are of great importance in the world.
Unfortunately the document needs REVISIONS.
Recommendations:
- The manuscript is suitable for publication __________
- The manuscript is suitable for publication after revisions. _XXXX___
- The manuscript is not suitable for publication____________
The manuscript suggested a reviewed meticulously, taking care of in addition to the made ones in the manuscript, the following observations and considerations:
1.- In order to make agile the arbitration of the present manuscript, I worked directly in the sent file. On the matter it is indicated that I am sending for you the file and Evaluation form, which are will have to open and to activate the tool CONTROL OF CHANGES in the program of Word for Windows.
2.- All the observations and commentaries to the manuscript were made directly in the mentioned file, reason why the authors will have meticulously to review each one of the proposed changes, respective observations and/or commentaries.
3.- In each one of the sections that compose the present manuscript they stand out in “red” the pertinent corrections.
4.- Check the summary due the modifications made.
5.- In the INTRODUCTION section modifications were made.
6.- In the section of MATERIALS AND METHODS: In general, the authors must remember that in this section the information with clarity will have to be provided, with detail, because some colleague could be interested in verifying the experiments of this information. Nevertheless, details about methods commonly used and some materials will be able to be omitted.
7.- In the section of RESULTS In this section, a greater description of the results is required: this is good for your article. However they (the results) has to be writing in the same form that you explain in your metodology.
It is also considered that the authors can include a section of Conclusions where in summarized form more and clear and mainly objective they present the conclusions with respect to the objectives raised in the introduction section.
8.-The section of REFERENCES: SEE COMENTS
Author Response
"Please see the attachment."

Reviewer 3 Report
Comments and Suggestions for Authors
This study showed the effect of drought on the morphological and physiological features of Wedelia trilobata, and W. chinensis.
There are many major and minor points that need to be clarified, please find the attached review file.

Author Response
"Please see the attachment."

Round 2
Reviewer 1 Report
Comments and Suggestions for Authors
I think that in the revised version of manuscript the Authors provide the most of necessary information and improvements.
I recommend that the paper can be published in Plants
Author Response
Thank you for your comments during the revision process; they have proven to be immensely helpful.
Reviewer 2 Report
Comments and Suggestions for Authors
congratulations
Author Response

(The authors gave the same response as above.)

Reviewer 3 Report
Comments and Suggestions for Authors
Still, there is an unsolved problem with the statistical analysis, especially with the comparative letters in the Figures and Tables, as follows:
1. In Figure 2. For example, in Figures 2A, 2C, and 2D, the letter A represents the highest values, while in Figure 2B, it represents the lowest.
2. In Figures 3A, 4A, and 4B, despite the obvious differences found, all treatments have the same letter (A).
3. In Figures 3B, 3C, and 3D, letter A represents the highest values.
4. Under Tables 1, and 2, the authors should explain the meaning of the letters, and the comparison is between each column or each row.
5. In the two Tables, the same problem concerning the arrangement of the letters, is letter A refers to the lowest or the highest values, please recheck and confirm.
6. By presenting the data concerning the phytohormones for each species in a separate Table, you cannot compare them statistically. I suggest presenting each hormone for both species in one Figure or Table. This will replace the two Tables with three Figures or Tables.
Author Response
"Please see the attachment."

Round 3
Reviewer 3 Report
Comments and Suggestions for Authors
1. The same problem in the statistical analysis still exists.
2. The authors must clarify if the comparison between means, in Tables, is between each column or each row. In either case, it is wrong.
3. The MS in this form cannot be accepted.
Round 4
Reviewer 3 Report
Comments and Suggestions for Authors
All comments were addressed by the authors, and the MS was markedly improved.